# Cardiac Toxicity Associated with Immune Checkpoint Inhibitors: A Systematic Review

**DOI:** 10.3390/ijms231810948

**Published:** 2022-09-19

**Authors:** Angela Cozma, Nicolae Dan Sporis, Andrada Luciana Lazar, Andrei Buruiana, Andreea Maria Ganea, Toma Vlad Malinescu, Bianca Mihaela Berechet, Adriana Fodor, Adela Viviana Sitar-Taut, Vasile Calin Vlad, Vasile Negrean, Olga Hilda Orasan

**Affiliations:** 1Department of Internal Medicine, Iuliu Hatieganu University of Medicine and Pharmacy, 400012 Cluj-Napoca, Romania; 2Department of Medical Oncology, Prof. Dr. I. Chiricuta Oncology Institute, 400015 Cluj-Napoca, Romania; 3Department of Dermatology, Iuliu Hatieganu University of Medicine and Pharmacy, 400012 Cluj-Napoca, Romania; 4Clinical Centre of Diabetes, Nutrition and Metabolic Disease, Iuliu Hatieganu University of Medicine and Pharmacy, 400012 Cluj-Napoca, Romania

**Keywords:** immune checkpoint inhibitors, myocarditis, pericarditis, arrythmia, cardiomyopathy

## Abstract

Immune checkpoint inhibitors (ICIs) are an important advancement in the field of cancer treatment, significantly improving the survival of patients with a series of advanced malignancies, like melanoma, non-small cell lung cancer (NSCLC), hepatocellular carcinoma (HCC), renal cell carcinoma (RCC), and Hodgkin lymphoma. ICIs act upon T lymphocytes and antigen-presenting cells, targeting programmed cell death protein 1 (PD1), programmed cell death protein ligand 1 (PD-L1), and cytotoxic T-lymphocyte antigen 4 (CTLA-4), breaking the immune tolerance of the T cells against malignant cells and enhancing the body’s own immune response. A variety of cardiac-adverse effects are associated with ICI-based treatment, including pericarditis, arrhythmias, cardiomyopathy, and acute coronary syndrome, with myocarditis being the most studied due to its often-unexpected onset and severity. Overall, Myocarditis is rare but presents an immune-related adverse event (irAE) that has a high fatality rate. Considering the rising number of oncological patients treated with ICIs and the severity of their potential adverse effects, a good understanding and continuous investigation of cardiac irAEs is of the utmost importance. This systematic review aimed to revise recent publications (between 2016–2022) on ICI-induced cardiac toxicities and highlight the therapeutical approach and evolution in the selected cases.

## 1. Introduction

Immune checkpoint inhibitors (ICIs) are an important advancement in the field of cancer treatment, significantly improving the survival of patients with a series of advanced malignancies. The most common types of malignancies that benefited from the introduction of ICI-based therapies are melanoma, non-small cell lung cancer (NSCLC), hepatocellular carcinoma, renal cell carcinoma (RCC), and Hodgkin lymphoma [1,2].

Normally, antigens are processed by antigen-presenting cells (APCs) and presented as peptide-major histocompatibility complexes (MHC) to naive lymphocytes in secondary lymphoid organs. This interaction is facilitated by the additional ligand-receptor binding of CD28 (expressed on T cells) and CD80 or CD86 (expressed on APCs). As a consequence, activated lymphocytes clonally proliferate and differentiate towards CD4^+^ helper or CD8^+^ cytotoxic phenotypes. These effector cells will further migrate and recognize the antigens in their original location via T cell receptors (TCRs). In physiologic conditions, several mechanisms dampen the activation of T-cells in order to prevent autoimmunity. Therefore, CTLA-4 is upregulated in activated T cells and negatively competes with CD28 in this process. Besides this, CTLA-4 is crucial for the maintenance of self-tolerance by the FoxP3^+^ regulatory T lymphocytes (Treg). Similarly, PD-1 expressed by activated T cells interacts with its ligand, PD-L1, which is expressed by APCs, lymphocytes, and epithelial cells in order to prevent aberrant immune reactions. In cancer, PD-L1 expression by tumors cells is a well-known mechanism of immune evasion. Blocking these pathways by use of ICIs restores the activity of the T cells but, as mentioned before, diminishes the immune tolerance and consequently promotes autoimmunity.

As ICI-based treatment is emerging in a wide spectrum of cancers with different primary sites such as gastro-intestinal (GI), head and neck, triple-negative breast cancer, and bladder cancers, it is of significant importance to identify and manage the adverse effects that are arising [2]. The lowered immune tolerance induced by ICIs brought to light a series of immune-related adverse events (irAEs) involving various organ systems.

IrAEs include dermatologic, GI, hepatic, endocrine, and other less common inflammatory events [1]. While most common irAEs can be managed with temporary immunosuppression using corticosteroids and/or other agents, some pose a vital risk for the patient and require prompt management and the complete discontinuation of ICI administration. IrAEs are classified according to the Common Terminology Criteria for Adverse Events from grades 1 to 5. Between 60 to 80% of patients treated with ICIs experience irAEs, and around 13–23% require permanent discontinuation as a result of grade 2, 3, or 4 AEs [1,3]. The risk and severity of irAEs is increased in patients treated with a combination of CTLA-4 and anti-PD1/PD-L1 [4]. A variety of cardiac adverse effects are associated with ICI-based treatment, including pericarditis, arrhythmias, cardiomyopathy, and acute coronary syndrome, with myocarditis being the most studied due to its often-unexpected onset and severity.

In this review, we have mainly focused on the clinical aspects of the ICI-related cardiac autoimmunity; a more detailed picture of these mechanisms is presented elsewhere [5,6]. Briefly, CD8^+^ lymphocytes recognize myocardial antigens, kill cardiomyocytes, and secrete IFN-γ. Further on, IFN-γ can induce PD-L1 expression in various cells, including cardiomyocytes and endothelial cells (ECs), as a protection mechanism. On the other hand, CD4^+^ lymphocytes are recruited and favor macrophage activation (Figure 1). It appears that the inhibition of the PD-1/PD-L1 and CTLA4 axes leads to cardiotoxicity in slightly different manners. For instance, in PD-1^−/−^ mice, cardiac troponin I autoantibodies are present and contribute to the development of autoimmune dilative cardiomyopathy [7], and PD-1-deficient MRL mice acquire lymphocytic myocarditis [8]. The immunization of PD-1^−/−^, but not PD-L1^−/−^, mice, with a peptide of the mouse α-myosin H chain, results in a primarily neutrophilic infiltration, also with an important lymphocytic component responsible for the secretion of IL-17A and IFN-γ [9]. Additionally, the transfer of PD-1^−/−^ CD8^+^ T cells in cMy-mOVA mice results in the higher proliferative activity of cytotoxic lymphocytes and elevated levels of cytokines and chemokines (IFN-γ, TNF-α, CXCL10, CCL3, CCL5) compared to their PD-1^+/+^ counterparts [9]. Compared to PD-1^−/−^ mice, CTLA-4^−/−^ mice do not exhibit autoantibodies, and a non-specific activation of CD4+ T cells may be responsible for the inflammatory response. Johnson et al. showed, by next generation sequencing, that myocardial and tumoral T cells resemble the same clonal origin [10], suggesting the existence of a shared epitope between the tumor and the myocardial cells. Thus, besides necrosis and oedema, histological samples of ICI-induced myocarditis consist of mononuclear cell infiltration: mixed CD4^+^ and CD8^+^ lymphocytic populations (usually with the predominance of cytotoxic T cells), macrophages (CD68^+^), and occasionally CD20^+^ B cells. Multinuclear giant cells were often encountered in severe, fulminant cases. Moreover, strong PD-L1 expression and the positive expression of HLA-DR (marker for IFN-γ production) are usually described. Notably, the absence of FoxP3^+^ Treg is also suggestive of ICI exposure [11,12]. Myocarditis is overall seen as a rare but frequently fatal irAE, with a fatality rate of up to 46% [13].

Regarding acute coronary syndromes, checkpoint blockade seems to be involved in the pathogenesis of atherosclerotic lesions, although the results are contradictory. Therefore, even if PD-1^−/−^ LDLR^−/−^ mice present an elevated formation of atherosclerosis plaques [14], a retrospective study showed that PD-1 inhibitor, nivolumab, was associated with a favorable course for atherosclerotic lesions [15].

Several novel checkpoint pathways are also currently under investigation. Lymphocyte activation gene-3 (LAG-3) binds to MHC-II and blocks antigen presentation, while T cell immunoglobulin-3 (TIM-3) and T cell immunoreceptor, with Ig and ITIM domains (TIGIT), are intrinsic inhibitors of T cells. In clinical settings, LAG-3 deficiency was shown to be associated with a proinflammatory state and a higher risk of coronary artery disease [16]. Similarly, TIGIT-positive Tregs possess a protective role regarding atherosclerotic lesions and are severely reduced in acute coronary syndrome [17]. However, TIM-3 expression was correlated with the severity of CHD, suggesting a different effect on lipid metabolism and atherosclerosis formation [18].

Considering the rising number of oncological patients treated with ICIs and the severity of the potential adverse effects, a good understanding and continuous investigation of cardiac irAEs is of the utmost importance. This systematic review aimed to revise recent publications published between January 2016–September 2022 on ICI-induced cardiac toxicities. Furthermore, we highlighted the treatment approach and evolution in selected cases.

## 2. Results

### 2.1. ICI-Induced Myocarditis

The most common cardiovascular side effect in ICI-treated oncologic patients is myocarditis. The inflammation of the heart muscle is often associated with arrhythmias, myositis, or myasthenia gravis (MG) (Table 1, the complete data regarding ICI-induced myocarditis is presented in the Appendix A). Thus, of the total studies included, 88 out of 128 (68.75%) were of case reports of ICI-associated myocarditis.

The majority of ICI-induced myocarditis cases were in male patients. At the same time, most of the cases were reported in subjects between 70 and 79 years, the youngest being 25 yo and the oldest 86 yo [19]. Regarding the underlying oncological disease, most of the cases were reported for patients diagnosed with melanoma [11,12,20,21,22,23,24,25,26,27,28,29,30,31,32,33,34,35,36,37,38,39], renal cancer [40,41,42,43,44,45,46,47,48,49], and lung neoplasms [33,50,51,52,53,54,55,56,57,58,59,60,61,62,63,64,65,66,67,68,69]

Most patients with ICI-induced myocarditis complain of shortness of breath, tachycardia or palpitations, fatigue, chest pain, cough, and episodes of syncope (See Table 1); cardiogenic shock was encountered in some of the cases [46,59,66]. It is noteworthy that a minority of patients had subclinical myocarditis [29,44,49,69]. However, even if they were asymptomatic from a cardiovascular standpoint, the presence of other irAEs, especially neuromuscular, guided further investigations. Notably, signs and symptoms of myositis/MG, such as diplopia, ptosis, blurred vision, dysphagia, muscle weakness, posterior neck pain, and neck drop, were reported in several studies (See Table 1). In such patients, dyspnea may be the result of both myocarditis-induced heart failure and myositis/MG.

In approximatively 25% of cases, ICI-induced myocarditis is a part of an overlap syndrome with MG and/or myositis [10,13]. A plausible explanation of this association is the presence of cross-antigens between the tumor and striated muscles [70,71]. Cancers with high-tumor burden (e.g., melanoma, NSCLC) display a higher incidence of irAEs, probably due to antigen spreading [72]. Although, in idiopathic MG, the association with myositis and/or myocarditis is relatively rare, the presence of anti-striational antibodies is correlated with a higher incidence of cardiac involvement [73,74]. Anti-AChR antibodies do not cross-react with heart muscles [75]. In severe myositis, troponin T could reflect the active regeneration of skeletal muscles [76].

Hepatitis, renal disturbances, thyroid dysfunction [28,30,57,77], Stevens–Johnson syndrome/toxic epidermal necrolysis [78], and DRESS syndrome [79] were also described in patients with ICI-induced myocarditis. Skin rashes and fever (See Table 1) were encountered in several patients.

Regarding the diagnosis, although a few patients presented a normal electrocardiogram (ECG) in spite of the cardiac symptoms (See Table 1), most of them presented electrocardiographic changes such as sinus tachycardia [30,80], bradycardia [24], atrial flutter [81] and atrial fibrillation [82], atrioventricular block [43,47,78], complete heart block [24,50,55], left bundle branch block [49], right bundle branch block (See Table 1), ST-segment depression [25,83], elevation [30,46,58,78] or non-specific ST-T changes [25,28], T wave inversion [77] ventricular tachycardia [26,59,84], and other changes, such as small QRS complexes [25,55].

Laboratory analyses, such as hs-TnI, CK, CK-MB, BNP, and NT-proBNP, were determined in the majority of cases (See Table 1). It is noteworthy that, when myositis/MG was suspected, complementary laboratory tests, such as anti-acetylcholine receptor antibodies (AChR), myohemoglobin, RyR-Ab, anti-NOR-90, anti-Ro-52 antibodies, anti-muscle specific kinase (MuSK) [85], anti-striated muscle, Asialo-GM1 [40], anti-GM1 [20], anti-SM [86], anti-TG, anti-TPO, anti-mitochondria, anti-SRP, and anti-PM/Scl100 [65] were carried out (See Table 1). In some patients, the EMG studies revealed a myogenic disorder [47] with a short duration and a reduced amplitude of muscle potential [80].

Normal wall motion and preserved EF were found in some of the cases (See Table 1); however, most patients had an abnormal echocardiogram. Thus, TTE revealed a depreciated EF (See Table 1), wall motion abnormalities, or global hypokinesis (See Table 1). Serial ultrasonographic examinations should be taken into consideration since, initially, normal kinetics or regional abnormalities can rapidly progress to global hypokinesis/akinesis [36,67]. Long-term screening is also needed in order to identify a possible evolution towards apical ballooning (Takotsubo cardiomyopathy) [55]. In terms of structural abnormalities, the echocardiogram depicted dilated atrial and ventricular cavities (See Table 1) and a thickening of the interventricular septum [38,49,83,87] and left ventricular wall, respectively [49,55,85,88]. Pericardial effusion [11,37,51,55], apical thrombi [55], and valvular regurgitation [38,83] were also found. Additionally, GLS measurements were performed in five patients, showing abnormal values in four of those patients (−14.4%; −15.7%) [11,20,39,49].

Chest X-rays uncovered an abnormal cardiothoracic index with pulmonary congestion [55,83], and pleural effusion [89]. The aforementioned abnormalities were also found with chest computer tomography (CT), which showed pleural and pericardial effusions, pulmonary oedema, and atelectasis [25]. Coronary angiographies ruled out acute coronary syndrome (See Table 1) in those patients with ICI-induced myocarditis.

Cardiac magnetic resonance imaging (cardiac MRI, CMR) was performed in the majority of cases. The most frequent findings associated with ICI-induced myocarditis are represented by oedema of the myocardium (T2-weighted imaging revealed high signal intensities consistent with acute myocarditis) and areas of late gadolinium enhancement (LGE), together with pericardial effusion (See Table 1). Extensive ventricular fibrosis was identified by MRI in some cases [46].

Cardiac catheterization and myocardial biopsies were performed on 23 patients. Endomyocardial biopsy (EMB) represents the gold standard procedure for myocarditis diagnosis [90]. The histopathology, together with the immunochemistry examinations, revealed an admixed interstitial inflammatory infiltration, consisting of T lymphocytes CD3+, CD8+, and macrophages (CD68+) (See Table 1). Furthermore, T lymphocytes expressed PD-1, while the CD68+ macrophages expressed PD-L1 checkpoint proteins [55,87]. The histopathological findings suggest that T lymphocytes play a crucial role in the pathogenesis of ICI-associated myocarditis. Additionally, several animal studies proved that PD-1 [8] and CTLA-4 [91] deficiencies predispose the animal to autoimmune myocarditis with a fatal outcome [8]. Interestingly, a recent study by Xia et al. showed that PD-L1-inhibitor treatment is responsible for the upregulation of M1 macrophage populations. An increased population of M1 macrophages has been documented in cardiac inflammation. Additionally, miR-34a expression was found to be higher in the macrophages of PD-L1 inhibitor-treated mice compared to the controls. The M1/M2 polarization of macrophages is regulated by miR-34a via KLF4 [92]. It is noteworthy that miR-34a exacerbates myocardial inflammation via the TGF-β/Smad-signaling pathway [93]. Collagen was also found admixed in with the inflammatory cells in a case reported by Norwood et al. Furthermore, the predominance of CD8+ with few CD4+ lymphocytes in the inflammatory infiltrate was once again established [28]. A case of chronic smoldering myocarditis was diagnosed when the EBM revealed myocardial infiltration by CD4+ and CD8+ T lymphocytes together with a considerable number of histiocytes and macrophages [46]. Neutrophils and eosinophils, together with myocardial necrosis [58,89], were also found in anatomopathological examinations. To better understand the pathophysiology and improve the diagnosis, several authors made use of immunohistochemistry (IHC) or cytokine analysis [55,87,94]. These could provide a better stratification of these entities and identify cell populations and cytokines in order to elect specific immunomodulatory therapies [11]. It is noteworthy that, in several patients, viral serology was positive for influenza [84], Coxsackie B [47], parvovirus B19, and HHV6 [64]. In these cases, it is not clear if the etiology of myocarditis was viral, autoimmune, or a combination of both.

#### 2.1.1. Treatment of ICI-Induced Myocarditis

The first therapeutic intervention in ICI myocarditis cases was the cessation of the incriminated drug. Steroid therapy is the main treatment for ICI-induced myocarditis. The efficacy of high doses of systemic corticosteroids, such as intravenous methylprednisolone, was documented in several studies (See Table 1). However, steroids could promote a transient myasthenic crisis in half of MG patients [49,95], and for that reason, it is recommended to associate it with intravenous immunoglobulin (IVIG) or plasmapheresis in these patients [96]. IVIG and plasmapheresis reduce autoantibodies in neuromuscular AEs [97]. Even if IVIG presents a relatively safer profile compared to other immunomodulatory strategies [94], it should be restricted in patients with high thromboembolic risk [85]. Additionally, plasma exchange (PLEX) is considered more suitable for MG patients with positive anti-MuSK antibodies [98].

In cases with an unfavorable evolution, the association of IVIG to the therapeutic regimen represents a reasonable option (See Table 1). The oral corticosteroid taper, following intravenous methylprednisolone administration, is mandatory (See Table 1). Other therapeutic interventions are represented by infliximab (monoclonal anti-tumor necrosis alpha antibody) [27,66,89,99], mycophenolate mofetil (MMF), together with oral prednisone [25], abatacept [47], or tacrolimus [30]. Furthermore, several patients do not respond to corticotherapy; in these patients, plasmapheresis, MMF, antithymocyte globulin, infliximab, alemtuzumab, tacrolimus, abatacept, tofacitinib, or tocilizumab, may be taken into consideration (See Table 1). In myasthenia-gravis, several alternatives include cyclophosphamide, azathioprine, methotrexate, or rituximab [63].

When a complete heart block is proved via ECG, a pacemaker (temporary or permanent) is required (See Table 1). In one case of hemophagocytic syndrome, cyclosporine was added to the therapeutic regimen [55]. PLEX [26] plasmapheresis [31,78], haemodialysis [87], non-invasive positive pressure ventilation [58], ECMO [30,66] intra-aortic balloon pump (IABP), and a defibrillator with cardiac resynchronization [66] were also needed in critically ill patients. Most of the patients required a complex therapeutic regimen consisting of a combination of the aforementioned interventions [29,31,78,89].

#### 2.1.2. Evolution of ICI-Induced Myocarditis

ICI myocarditis was lethal in 23 of the cases included in the present review. An anti-PD-1 antibody was administered in most of the fatal cases, namely pembrolizumab [55,89,99] or nivolumab [45,59,87]. Fatal outcomes saw myocarditis patients treated with an association of ICI (anti-PD-1 + anti-CTLA-4 inhibitors) [27,31]. In the present review, myocarditis onset presented large temporal variations in relation to ICI administration. Thus, it developed as early as 1 week after the first ipilimumab + nivolumab infusion [29], but it was also diagnosed after long-lasting treatment with pembrolizumab (8 cycles, 1 year) [57].

It is noteworthy that there is a lack of medical history documentation in the majority of ICI-associated myocarditis patients. Special consideration should have been attributed to the previously diagnosed cardiovascular pathologies and autoimmune diseases. Another critical aspect that should be cautiously taken into consideration is represented by the infectious ethology of myocarditis. Additionally, several patients had a history of thymoma (See Table 1). The association between thymoma, myocarditis [100], MG, and myositis [101,102] is well known. Thus, thymoma-induced autoimmunity is also plausible in patients with thymic tumors treated with ICI [35].

### 2.2. ICI-Induced Pericarditis

ICIs have been associated with the development of other various cardiovascular toxicities, including pericarditis. Regarding pericardial diseases associated with ICI-based therapy, we analyzed several case reports and the time to onset of pericarditis was 5 to 340 days; most of the cases were reported in male subjects with an average age of 60 years. Moreover, at particular intervals of time, the patients developed clinical manifestations, like dyspnea, thoracic pain, fatigue, and general malaise (See Table 2). Furthermore, a patient presented dyspnea with tachycardia and a low-grade fever [103], limb oedema, and increased body weight [104], while others developed severe hypotension due to cardiogenic shock [105]. With regard to the associated comorbidities with ICI-related pericarditis, only asthma and hypertension were mentioned [106]. It is also noteworthy that previous immune toxicity, characterized by thyroiditis [103,107], autoimmune hepatitis [104,107], generalized pruriginous rash, [106,107], colitis, and MG [106], may show a predisposition to autoimmune toxicity with these drugs.

After the initial clinical evaluation, for a complete diagnosis, the management of ICI-related pericarditis included an ECG, echocardiogram, CT, and MRI. The ECG findings ranged from normal ECGs [103,104,108,109] to new T-wave inversions in anterolateral leads and prolonged QT intervals [107,110], sinus tachycardia and decreased QRS voltage [105,107], atrial fibrillation with a high heart rate and an ST segment elevation [96,111], and in some cases, non-specific repolarisation anomalies [106].

Most of the patients with ICI-induced pericarditis experienced preserved LVEF and pericardial effusion without signs of hemodynamic impairment [104,105,106,107,108,109,111], but there were two cases described in which the echocardiogram emphasized an important restrictive pericardial effusion with “swimming heart”, suggestive of cardiac tamponade [103,110].

A single chest X-ray showed moderate bilateral pleural effusions and an enlarged heard silhouette [107]; in most of the cases, for better accuracy, CT scans were performed, which confirmed the pericardial effusion [103,104,107,109]. For a better investigation, the cardiac MRI showed a marked pericardial hyperintensity in T2-weighted sequences as a sign of active inflammation [109]. Cardiac MRI proved the presence of pericardial thickening, although without evidence of myocardial oedema or marked late godalinium enhancement in the pericardium [104].

What is more, the cardiac enzymes, such as troponin levels and BNP, remained within the normal range in patients with ICI-related pericarditis [105,107,110,111], except for one case where the serum troponin-T level was high, but there was no evidence of active myocarditis from the endomyocardial biopsy. Elevated serum troponin indicated myocardial inflammation that was associated with severe inflammation of the pericardium [104]. In some cases of ICI-pericarditis, the histopathologic evaluation showed only non-specific chronic inflammation with extensive fibrosis and lymphocyte infiltration, with no evidence of malignancy or microorganisms [103,105,107].

Regarding the treatment of ICI, pericardiocentesis remains the preferred procedure for the acute management of massive pericardial effusions by either mechanism [103,104,105,107,108,110]. Corticosteroids are a prefered treatment, needing only a prompt initiation. [96] In our reviewed cases, corticosteroids were administered to all the patients with ICI-related pericardial effusions (See Table 2). The treatment starts with high doses of prednisone (1–2 mg/kg) with a tapering off of at least 4 weeks in selected cases. Transplant rejection doses must be administered in unresponsive cases, which means 1 g methylprednisolone given daily in association with mycophenolate mofetil, infliximab [104], or anti thymocyte globulin [110].

In the present review, the majority of ICI-induced pericarditis has not developed any other immune-related adverse events (See Table 2). These case reports highlight the importance of prompt diagnosis and high-dose corticosteroid treatment for immune-related pericarditis. Nevertheless, in one case (of a 54-year-old patient diagnosed with lung adenocarcinoma who was treated with ICI), it was discovered that the patient had moderate paraneoplastic pericarditis; a tapering scheme of prednisone was administered (10 mg/day, every 2 weeks) in combination with colchicine for 6 months to prevent another pericardial effusion. His cardiac condition was evaluated again by echocardiogram, which was substantially stable compared to the previous one. Thus, a second cycle of chemotherapy was initiated without administering pembrolizumab (given the high doses of prednisone ongoing). Unfortunately, the patient experienced acute dyspnea and succumbed to sudden death after less than two months after the first cycle of chemotherapy. No autopsy was conducted as per the family’s wish [111].

To stratify the patient risk before starting ICI treatment, it is recommended that complete blood tests be performed, including troponin and brain natriuretic peptide (BNP), ECG, and echocardiogram, besides the usual clinical follow-up. The echocardiogram, cardiac biomarkers, and/or cardiac MRI are useful investigations for diagnosing late-onset cardiotoxicity, both in the initial and later phases of ICI therapy [96].

### 2.3. ICI-Induced Arrythmias

New-onset or pre-existing arrhythmias with ICI-use (Table 3) ranged from tachyarrhythmias, such as atrial fibrillation [112] and sinus tachycardia [113] to bradyarrhythmias, such as atrioventricular block [114]. These adverse effects were observed mostly in male patients with metastatic melanoma [112,115,116], metastatic sarcoma [113], metastatic NSCLC [115,117], and end-stage liver cancer [118].

After a period (running from two weeks to five months) following the initiation of ICI therapy, the patients met the definition of ICI-induced arrhythmias. They tended to be older and more likely to have a history of hypertension, diabetes mellitus [112,115,117], and hyperlipidemia [117]. It is noteworthy that a single patient with pre-existing atrial fibrillation underwent three prior ablations in the past [112]. Regarding the anamnesis and clinical examination in patients with arrhythmias, some patients remained asymptomatic [112,116]. Most of them developed manifestations, like increased fatigue, generalized malaise, dizziness, or syncope [112,113,118,119], while others experienced palpitations [119] or worsening dyspnea [115].

A combination of ECGs, echocardiograms, cardiac catheterizations, and biomarkers was used to identify cases of ICI-induced arrhythmias after the initiation of immune checkpoint inhibitor therapy. Twelve-lead ECGs were recorded with the presence of atrial fibrillation with rapid ventricular response [112], sinus tachycardia associated with first-degree atrioventricular block, right bundle branch block, and left anterior fascicular block, then accelerated junctional rhythm with complete heart block [113], symptomatic sinus bradycardia of 40 bpm [112,118], a new first-degree AV block [116], Mobitz type 2 s-degree AV block of 30 bpm followed by complete heart block with a ventricular rate of 22 bpm after 3 h [117], a third-degree AV block with a bradycardia of 44 bpm [115], and complete AV block with wide QRS complexes [119].

Furthermore, QT dispersion (QTd) is measured as the difference between the longest and the shortest QT intervals within a 12-lead ECG. Prolongation of corrected QT interval (QTc) is an important indicator of arrhythmogenesis [116]. In this review, there was one study that mentioned a significant increase in QTd after treatment initiation with a combination of ICI, while no significant difference was observed in patients treated with PD-1i monotherapy [116].

In 90% of the cases with ICI-induced arrhythmias, the transthoracic echocardiogram did not reveal abnormalities in wall motion, with the preservation of the ejection fraction of the left ventricle. Nevertheless, an unusual case of progressive conduction abnormalities after the initiation of checkpoint inhibitors, anti-PD-L1 and anti-CTL4, showed a moderate decrease in the left ventricle ejection fraction (30–35%), associated with kinetic disorders (anteroseptal hypokinesis and apical akinesis with septal motion consistent with conduction abnormality) in the setting of a normally sized left ventricle. The systolic function and contractility of the right ventricle were reduced in spite of no significant valvular disease or pericardial effusion [113].

Laboratory results were negative for elevations in troponin or BNP in half of the cases [116,117,118], but increased myoglobin, creatin-kinase, and troponin were detected in the rest of them [113,115,119]. Cardiac catheterization was also performed in three cases of ICI-induced arrhythmias, which did not reveal any significant stenosis [115,119] or there were minor luminal irregularities in the coronary arteries [113].

The evaluation for the immune-related effects, like hyperthyroidism, was notable in only one case, which was felt to possibly be drug-induced toxicity from the ICIs, treated with methimazole and steroids [112]. A suggested approach to the management of a patient with suspected ICI-induced arrhythmias is outlined following the guidelines for the management of patients presenting different arrhythmias. Self-limited AF resolved spontaneously after about 48 h [112], persistent AF managed with an echocardiogram (TEE)-guided cardioversion, anticoagulation and beta blockers and antiarrhythmic drugs [112], implantation of a temporary pacemaker, then permanent pacemaker for the heart conduction disorders [113,115,119] and association of cortisone in immune-mediated sinoatrial node dysfunction with immediate improvement of symptoms and bradycardia without relapsing-symptoms [118]

None of the ICI-induced arrhythmias cases required stopping treatment with the ICIs, and in most instances, the arrhythmias could be managed in a relatively straightforward manner with positive evolution, except for one case where pembrolizumab was discontinued due to the progressive status of his HCC. Unfortunately, one patient 26 days after beginning anti-PD-1treatment. Considering the metastatic malignant underlying disease, and the worsening hemodynamic and respiratory situation, the relatives made the decision to reduce medical care with an unfavorable prognosis.

### 2.4. Takotsubo Cardiomyopathy and Acute Heart Failure

ICIs have also been associated with the development of other various cardiovascular toxicities, such as acute heart failure, cardiomyopathies, Takotsubo syndrome, and acute coronary syndromes. Concerning cardiomyopathies, including Takotsubo syndrome, we analyzed several case reports and the time to onset of cardiomyopathy is 4 to 112 days; most of the cases were reported in male subjects with an average age of 64 yo, the youngest being 42 yo and the oldest 79 yo.

Regarding the underlying oncological disease, cases have been reported in patients diagnosed with melanoma [120,121,122,123,124], lung neoplasms [125,126], hepatocellular carcinoma [127], breast cancer [128], and squamous cell carcinoma of the lip [129]. Most of the cases were described as Takotsubo cardiomyopathies [120,121,122,123,127,129], while two cases of acute heart failure were also described [125,126]. It is noteworthy that, in two cases, the ICI-induced cardiomyopathies were associated with acute renal failure [129] and with myopericarditis, respectively [127]. The majority of patients with ICI-induced cardiomyopathies complained of chest pain and shortness of breath. Diaphoresis, nausea, vomiting, and lower extremity oedema were encountered in some of the cases (Table 4).

In regard to the diagnosis, most of the patients presented ECG changes, such as sinus tachycardia [123], atrial flutter [126], ST elevation [121,126], or T wave changes [120,122]. A TTE was performed in all the cases and was always abnormal. In patients diagnosed with Takotsubo syndrome, TTE revealed apical akinesis with apical ballooning, hyperdynamic basal LV segments, and a reduced cardiac ejection fraction [120,121,122,127,129]. One case of atypical Takotsubo cardiomyopathy was also described whereby the TTE revealed hypokinesia of the mid inferior and mid anterior septum, mid inferolateral, mid anterolateral, mid inferior and mid anterior wall, sparing the apical and basal segments [123]. In acute heart failure cases, the wall motion abnormalities were not systematized [125,126].

Chest X-rays uncovered pulmonary congestion, pulmonary effusion, and cardiac dilation, but these were performed only in a few cases [123,126]. A CMR was obtained in the majority of cases, mainly to rule out the diagnosis of ICI-induced myocarditis. It is noteworthy that one patient had concomitant myopericarditis and Takotsubo syndrome following ICI therapy. In this case, the CMR revealed mid-myocardial to subepicardial delayed enhancement (MDE) in the inferior lateral walls, as well as in the apex [127].

The laboratory analyses (TnI, CK, CK-MB, BNP, and NT-proBNP) were determined in the majority of cases and were almost always elevated, which raised the question of an acute coronary syndrome [120,121,122,123,125,127,129]. Subsequently, a coronary angiography was performed in the majority of the cases and an acute myocardial infarction was ruled out [121,122,123,125,126,127,129]. Cardiac catheterization and myocardial biopsies were performed in two patients, which excluded a possible myocarditis [120,126].

The first therapeutic intervention (when ICI-induced side effects were suspected) was the cessation of the incriminated drug. Immunosuppressive therapy with intravenous corticosteroids was initiated and continued with oral prednisone taper when myocarditis was diagnosed [127] or suspected [122,125,127,129]. In all cases, guideline-directed heart failure treatment was started with betablockers, ACE inhibitors, and diuretics, as well as IV dopamine and tolvaptan in one severe acute heart failure case [126]. After the ICI treatment was discontinued and heart failure treatment was initiated, the evolution was favorable in all the cases, with an improvement in the symptoms and the left ventricular ejection fraction [120,121,122,123,125,126,127,129].

### 2.5. Acute Coronary Syndrome

We analyzed the case reports of acute coronary syndromes (ACS) related to ICI administration, and the time to onset of ACS is 2–365 days, and most of the cases were reported in male subjects with an average age of 67 years, the youngest being 52 yo and the oldest 87 yo. Concerning underlying oncological disease, the cases were reported in patients diagnosed with lung cancer [130,131,132,133,134], melanoma [135], and renal cell carcinoma [136], as we have presented in Table 5.

More than 70% of the cases were described as non-ST elevation myocardial infarction (NSTEMI), but ST-elevation myocardial infarction (STEMI) [133] and vasospastic angina were also described [136,137,138]. The main clinical manifestations of ICI-associated ACS were chest pain and dyspnea; only one case reported no symptoms, and the diagnosis was based on the elevated cardiac biomarkers [131]. Concerning concomitant autoimmune diseases, one patient developed grade five colitis while being hospitalized for ICI-related myocardial infarction [134]. Another patient presented related concomitant autoimmune manifestations, namely erythema multiforme, thyroid dysfunction, and interstitial pneumonia [131], while another case with associated ICI-induced hepatitis was also reported [131]. The concomitant autoimmune diseases were treated with high-dose intravenous corticosteroids, followed by oral corticosteroids taper.

Most of the patients had an abnormal ECG with ST-depression in various territories [130,135,136], T-wave inversions [132], Q-waves [116,134] and ST-elevation [132]. One case report revealed no ECG abnormalities [131]. The laboratory analyses (TnI, CK, CK-MB, BNP and NT-proBNP) were determined in the majority of cases and revealed abnormal values, particularly the troponins [130,131,132,133,134,135,136]. In every case, emergency coronary angiography was performed and revealed significant stenosis or occlusion of the coronary arteries [130,131,132,133,134,135], except for the case of vasospastic angina [136].

Most of the patients underwent a percutaneous coronary intervention with implantation of drug-eluting stents [112,113,114,115,116]. However, one patient with underlying severe CAD was surgically treated with coronary artery bypass grafts (CABG) for the affected arteries [135]. The patient diagnosed with vasospastic angina was treated with calcium channel blockers and nitrates, and the symptoms gradually disappeared [136]. All the patients were discharged with chronic myocardial infarction treatment, namely dual antiplatelet therapy, ACE inhibitors, beta-blockers, and statins, except for the patient diagnosed with vasospastic angina [136]. The evolution of the patients was almost always favorable [130,131,132,133,134,135,136], except for one patient who died 16 days after stenting due to refractory shock without ACS recurrence [134].

### 2.6. IrAEs Reported in Clinical Trials

Between 2017 and January 2022, there were published trials that reported cardiac irAEs following the administration of ICIs, the most common AE being myocarditis. Maio et al. [139] conducted a double-blind placebo-controlled trial, evaluating the use of tremelimumab as second- or third-line treatment in patients with relapsed malignant melanoma, reporting a multitude of cardiac AE. In the treatment group, one case of myocardial infarction (vs. none in the placebo group) was identified, along with one case of cardiac failure (vs. one in the placebo group) and twelve cases of pericardial effusion (vs. six in the placebo group). A dose-finding and dose-expansion phase 1b trial conducted by Choueiri et al. [140], regarding the use of avelumab, a PD-L1 inhibitor, in association with axitinib, a tyrosine kinase inhibitor (TKi), in patients with clear-cell renal-cell carcinoma reported one case of myocarditis diagnosed before the first evaluation of the patients, resulting in the death of the patient. Among the six deaths reported in this study, this was the only one caused by irAEs, with the other five being a result of disease progression. Juergens et al. [141] assessed the efficacy of durvalumab with or without tremelimumab in combination with platinum-doublet based chemotherapy in several types of malignancies, the most common being NSCLC. There was one reported case of myocarditis, which was confirmed using post-mortem. The ESCORT trial assessed the efficacy of camrelizumab vs. the investigator’s choice chemotherapy regime in advanced or metastatic esophageal squamous cell carcinoma. A total of 10 treatment-related deaths were reported, 7 of them for camrelizumab (one with myocarditis) [142].

Hasson et al. [143] assessed the safety of reintroducing immunotherapy in patients that survived ICI-induced myocarditis. Three patients were chosen from a total of seven and ICIs were reintroduced under close surveillance and in combination with low dose PO prednisone and cardio-protective therapy (angiotensin-converting-enzyme inhibitors, beta-blockers, or mineralocorticoid receptor antagonists). One of the three patients exhibited heart failure symptoms which required the cessation of the ICI regiment, while the other two continued treatment with durvalumab and pembrolizumab without irAEs at the time of data cut-off.

#### Limitations of the Study

For the moment, a lot of information is emerging from daily clinical practice but must be treated with caution. Considering the lack of full data about patients’ profiles (regarding their cardiovascular and non-cardiovascular history and associated risk factors), further research is needed to fully assess the relationship between ICI and cardiotoxicity.

## 3. Material and Methods

We reviewed all publications from PubMed, Embase, and Medline databases using the terms: (“immunotherapy”[tw] OR “CTLA4”[tw] OR “PD1”[tw] OR “PDL1”[tw] OR “PD-1”[tw] OR “PD-L1”[tw] OR “Immune checkpoint inhibitors”[tw]) AND (“cardiotoxicity”[tw] OR “myocarditis”[tw] OR “Myocardial Infarction”[tw] OR “Heart Failure”[tw] OR “Acute Coronary Syndrome”[tw] OR “Arrhythmias, Cardiac”[Mesh] OR “Takotsubo Cardiomyopathy”[tw] OR “Pericarditis”[tw] OR “Coronary Vasospasm”[tw] OR “Shock, Cardiogenic”[tw] OR “Heart Conduction System”[tw]) NOT (“review”[tiab] OR “meta-analysis”[tiab]). Only human studies from the past five years were selected for screening, including case reports/case series. Duplicates were removed and only articles written in English with the presence of an abstract were reviewed. Reviews, systematic reviews, animal, and in vitro studies were excluded (Figure 2).

## 4. Conclusions

ICI-induced cardiac toxicity comprises a large field of morphological and functional abnormalities, including myocarditis, pericarditis, Takotsubo cardiomyopathy, and acute coronary syndrome. Despite only a few reports from the first clinical trials involving ICI, many cases were documented in later years. For this review, we identified and collected relevant data from the last five years in the literature in order to highlight the clinical and paraclinical patterns, as well as any therapeutic approaches. Although rare, these adverse events can lead to fatal outcomes, especially in the case of myocarditis. The precise mechanisms are not yet fully elucidated, and consequently, the treatment lacks standardization. Further research is needed for the better stratification of these pathological entities and the proper design for early diagnosis and treatment strategies.

## Figures and Tables

**Figure 1 ijms-23-10948-f001:**
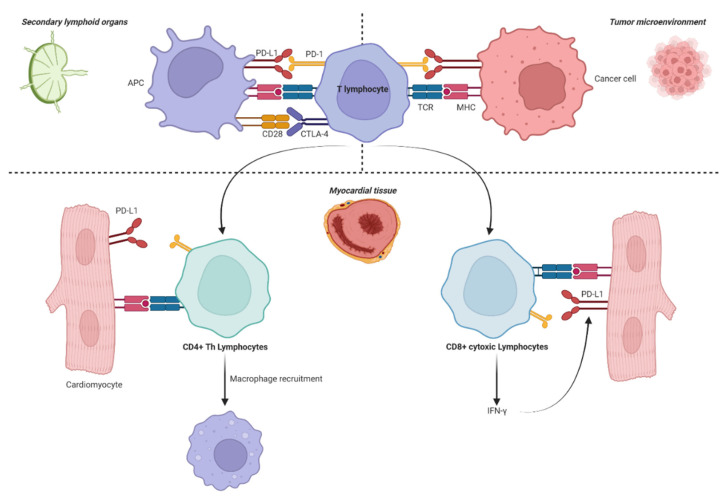
The CD8+, CD4+ T cells, and macrophages interplay within the pathogenesis of ICI-related cardiotoxicity.

**Figure 2 ijms-23-10948-f002:**
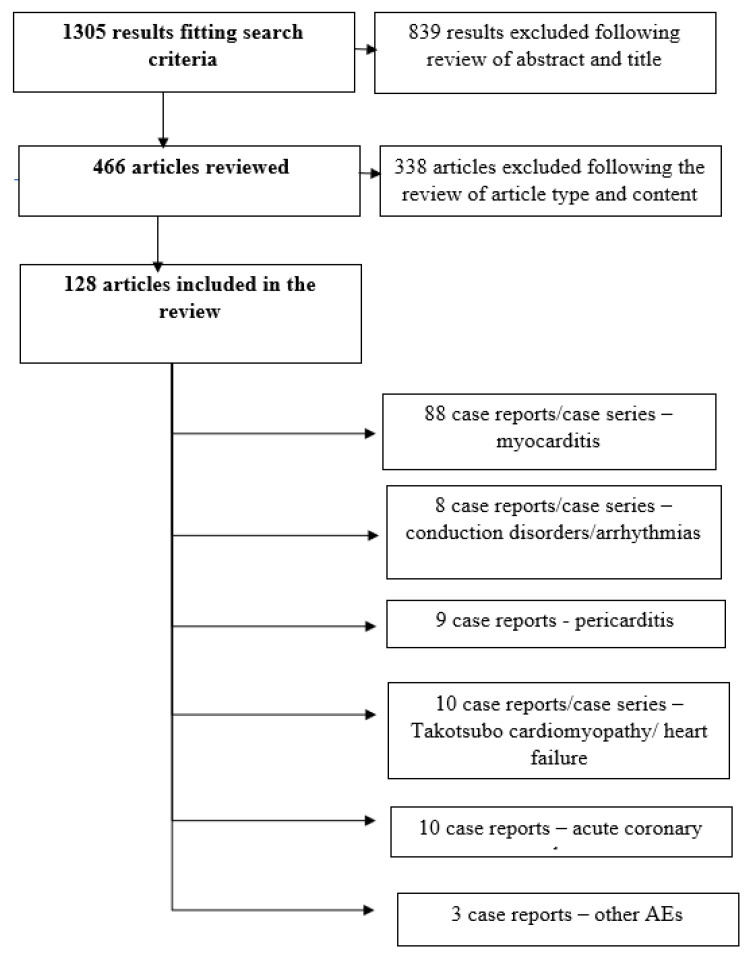
Flow diagram of the paper selection processes.

**Table 1 ijms-23-10948-t001:** Case reports published from 2021–2022—ICI-related myocarditis.

Study	Patient Characteristics	Medical History	Symptoms	Diagnosis	CV Side Effect	ICI	Type of Cancer	Myocarditis Onset	Myocarditis Treatment	Evolution
Ida, 2022	♀, 81	HBP dyslipidemia	high-grade fever whole-body rash altered consciousness	↑CK; ↑CKMB, troponin I, C-reactive protein, ECG, TTE CMR endomyocardial biopsy	Myocarditis	PD-L1i + PD-1i	advanced melanoma,	1 week	Methylprednisolone followed by prednisolone IGIV	favorable
Nguyen, 2022	♂, 25	N/A	chest pain, subtle myalgia	Coronary angiography cardiac MRI endomyocardial biopsy Troponin-T creatine-kinase	Myocarditis myositis	PD-1i	hymoma	2 weeks	Methylprednisolone mycophenolate-mofetil loading dose of intravenous abatacept oral ruxolitinib	cardiogenic shock VT extracorporeal life support On day 40, the patient fully recovered clinically
Okauchi, 2022	♂, 60	Smoking history	Cough, dyspnea	Chest RX BNP TTE	Myocarditis	PD-1i	Squamous cell carcinoma	130 weeks from initiation	Diuretics, beta-blocker	Favorable
Zhou, 2022	♂, 67	N/A	fever, chest pain and dyspnea	Chest computed tomography, TTE, ECG, BNP, troponin T, CK	myocarditis	PD-L1i	lung squamous cell carcinoma stage IV	days after lastcycle	methylprednisolone	favorable
Zhao, 2022	♂, 60	N/A	fever, tachycardia, hypotension, fatigue, dyspnea	ECG, TTE	Myocarditis + hypothyroidism	PD-1i	soft tissue sarcoma	8 weeks	IV mPSL	favorable
Lorente-Ros, 2022	♂, 70	nephrectomy	2 episodes in the previous 12 h of severe dizziness, dyspnea, and profuse sweating.	ECG troponin I,C- reactive protein, TTE ECG- 3rd AVB, coronary angiography, brain CT autoimmunity lab tests, brain, MRI, lumbar puncture, EEG	Myocarditis + Encephalitis	PD-1i + CTLA-4i	renal cell carcinoma	19 days	Temporary pacemaker high-dose iv corticosteroids intravenous immunoglobulins	delirium deterioration in his level of consciousness, intubation. Extubate -> reintubation discharged
Saishu 2022	♀, 55	N/A	quadrantanopia, ocular motility disorder, diplopia, dysphagia, ocular motility disorder, muscle weakness of the extremities, bilateral ptosis muscle weakness.	↑CK; ECG, TTE; anti-AchR ab.	Myocarditis, myositis/MG	PD-1i	mela- noma	2 weeks	IGIV, prednisolone Intubation for MV, mPSL plasma exchange Tracheostomy	favorable
Yang, 2022	♀, 51	contrast agent allergy	high fever, mild dyspnea, and systemic rash.	Liver function indexes cardiac markers CT examination	myocarditis hepatitis	PD-1i	breast cancer (TNBC)	3 days	Iv methylprednisolone Antibiotics hepatic protectors	favorable
Ederhy, 2021	♀, 60–70 approx.	N/A	diplopia	↑TnI; ECG; CMR, coronary angiography, EMB	Myocarditis (subclinical)	PD-1i	Lung cancer (unmentioned type, metastatic)	3 infusions + 10 days	steroids, plasmapheresis	favorable
Tsuruda, 2021	♂, 75	N/A	asymptomatic	↑cTnT, CK, CK-MB; ECG, Echocardiogram, CMR, EMB	Myocarditis (subclinical), TTS	PD-1i	NSCLS (squamous, recurrent)	3 weeks	mPSL	fatal
	♂, 47	N/A	asymptomatic	↑cTnT, CK, CK-MB; CMR	Myocarditis (subclinical)	PD-1i	Ethmoid sinus cancer	3 infusions + 16 days	mPSL; IVIG (progressive thrombocytopenia); cyclosporine (hemophagocytic syndrome)	favorable
	♂, 63	N/A	hypotension (84/42 mmHg), tachycardia (132 bpm), tachypnea (22 rpm); high fever, decreased appetite,	↑cTnT, CK, AST, ALT, CRP, Cr; ↓WBC, Hb, PLT; ECG, Echocardiogram	cardiac complication of cytokine-releasing syndrome	PD-1i	Hypopharyngeal cancer	5 infusions + 32 days	cardioversion, extracorporeal hemoperfusion with polymyxin B + continuous hemodiafiltration, catechocardiographylamines, broad-spectrum antibiotics, recombinant thrombomodulin, IVIG, high-dose corticosteroids	favorable
Tanabe, 2021	♂, 75	N/A	posterior neck pain, neck drop	↑TnI, CK, CK-MB; ↑Eo (834/μL), ↓eRFG; DLST (+); Echocardiogram, coronary angiography, CMR,	Myocarditis (subclinical)	PD-1i + CTLA-4i	RCC (clear cell, metastatic)	53 days	prednisolone	favorable
Barham, 2021	♀, 79	N/A	dizziness, abdominal bloating, hypoxic	↑LDH; ECG; EMB	Myocarditis (grade 4); hyperprogression	PD-1i + CTLA-4i	Melanoma (vaginal, metastatic)	23 days	steroids; carboplatin + paclitaxel (salvage therapy), atropine, pacemaker (for AVB III)	fatal
Xie, 2021	♂, 67	N/A	exertional dyspnea, ptosis, blurred vision, quadriparesis	↑TnI, CK, CK-MB, AST, ALT, BNP, Mb; ECG; Echocardiogram; coronary angiography	Myocarditis (fulminant), MG crisis, hepatic dysfunction; delayed ir pneumonitis	PD-1i + pemetrexed + carboplatin	LCNEC (metastatic)	2 weeks	mPSL; pacemaker (temporary permanent); ganciclovir/cefmetazole	favorable
Hu, 2021	♂, 63	N/A	chest tightness, limb weakness, dorsal myasthenia, diplopia, dysphagia	↑Hs-TnI, CK-MB, NT-proBNP, CK; Echocardiogram, CMR; anti-β1AR ab, CC ab, anti-myosin heavy chain ab, ribonucleoprotein ab	Myocarditis + MG	PD-1i	ureteral urothelial cancer IV	3 weeks	mPSL IVIG	favorable
Wintersperger, 2021	♂, 52	N/A	fatigue dyspnea	↑hsTnI, CK, BNP; ECG, Chest CT, Echocardiogram, CMR, EMB	Myocarditis	PD-L1i + investigational ICI	melanoma	3 weeks	mPSL prednisone infliximab IV MMF	favorable
	♀, 60	N/A	general- ized weakness muscle pain fatigue fever	↑CK, hsTnI; ECG, Coronary angiography, CMR, EMB	Myocarditis	PD-L1i	gynecological cancer	2 weeks	mPSL prednisone	favorable
	♀, 49	N/A	fever cough	↑hsTnI, BNP; chest CT, ECG, CMR	Myocarditis	PD-L1i	triple-negative breast cance	2 weeks	MMF prednisone	favorable
	♀, 74	N/A	general pain, progressive muscle weakness diplopia	↑hsTnI, BNP, ECG, Coronary angiography, CMR	Myocarditis	PD-L1i	gynecological cancer	2 weeks	mPSL prednisone MMF	favorable
Stein-Merlob, 2021	♀, 60	N/A	palpitations reduced exercise tolerance, cool extremities altered mental status	↑Tn, BNP; ECG, Echocardiogram, Coronary angiography, CMR	Myocarditis Ocular myasthenia, Colitis hepatitis	PD-1i	Colon cancer		Metoprolol succinate, lisinopril Continued immunosuppression spironolactone, Oral amiodarone, wearable defibrillator.dopamine Nitroprusside milrinone, VA-ECMO	favorable
Shen, 2021	♀, 53	N/A	cough chest congestion, muscle weakness fatigability drooping eyelids,	↑CK, CK-MB; ECG	Myocarditis, hepatitis, renal dysfunction, hypothyroidism	PD-1i + paclitaxel + platinum	type B3 thymoma	3 weeks	Magnesium isoglycyrrhizinate reduced glutathione injections, prednisone mPSL euthyrox pyridostigmine	favorable
Miyauchi, 2021	♂, 71	hypertension, DM2, hyperuricemia	Asymptomatic, chest tightness, shortness of breath, cardiogenic shock	↑CK, CM-MB, TnI, NT-proBNP; ECG, catheterization, EMB, CMR	Myocarditis	CTLA-4i + PD-1i	RCC	8 weeks	dopamine, dobutamine, noradrenaline intra-aortic balloon pump was inserted, adaptive servo ventilation mPSL prednisolone	favorable
Luo, 2021	♀, 47	N/A	diplopia, myalgia, limb weakness, dysphagia, dyspnea	↑TnI, CK, ECG, EMG; RyR-ab, AChR-ab, anti-fibrillarin ab, anti-NOR-90 ab anti-Ro-52 ab	Myocarditis, myositis, MG	PD-1i	thymoma	3 weeks	neostigmine IVIG mPSL prednisolone pacemaker	favorable
Li, 2021	♂, 62	hypertension, coronary heart disease	fever lethargy, cognitive dysfunction tachypnea hypoxia hypotension oliguria	↑Mb, Tn, CK-MB; ECG	cardiotoxicity kidney toxicity.	PD-1i	lung adenocarcinoma	48 weeks	mPSL continuous renal replacement therapy	favorable
Jespersen, 2021	♂, 57	N/A	headache myalgia, palpitations binocular diplopia, ptosis, muscle weakness	↑TnI, CK-MB, Mb, CK; ECG, EMG, Echocardiogram, CMR, AChR-ab,	Myocarditis + myositis	CTLA-4i + PD-1i	RCC	2 weeks	temporary pacemaker. mPSL abatacept MMF implantable cardio-defibrilator	favorable
Iwasaki, 2021	♀, 70	hypertension, aortic stenosis, chronic renal failure	shortness of breath fatigue	↑CK, CK-MB, TnT, NT-proBNP; ECG, Echocardiogram, CMR, Cardiac catheterization, EBMCoronary angiography	Myocarditis + myositis	PD-L1i	HCC	<1 week	cariperitide, furosemide, mPSL prednisolone	favorable
Hernández, 2021	♀, 48	N/A	shortness of breath, dyspnea, bilateral ptosis blurred vision	↑Hs-TnI, NT-proBNP, CRP, CK; ECG, Echocardiogram, Coronary angiography, EMB; AChR-ab	Myocarditis + myositis (MG)	PD-1i	thymoma	<2 weeks	IV isoproterenol drip mPSL Infliximab temporary pacemaker, dual-chamber pacemaker. intravenous amiodarone, noradrenaline dobutamine, intravenous anti-thymocyte globuline, pyridostigmine, ECMO	fatal
Giblin, 2021	♀, 47	N/A	dermatitisdiarrhea, palpitations	↑hS-TnI, BNP, Echocardiogram, CMR, Coronary angiography, EMB	Myocarditis (subclinical)	CTLA-4i + PD-1i	melanoma	1 week	mPSL prednisolon, IVIG	favorable
Cao, 2021	♂, 69	N/A	ptosis, diplopia, shortness of breath,	↑CK, CK-MB, Mb, hs-TnT, NT-proBNP, LDH, ECG; Echocardiogram, EMG	Myositis, Myocarditis SJS/TEN	PD-1i	esophagogastric junction carcinoma	2 weeks	mPSL IVIG plasmapheresis	favorable
Ai, 2021	♂, 72	N/A	asymptomatic	ECG; CMR	Myocarditis (DRESS)	PD-1i	gastric adenocarcinoma	3 weeks	SCS (for DRESS)	

Abbreviations: ↑, increase; ↓, decrease; Ab, antibody; ACEi, angiotensin-converting-enzyme inhibitors, AchR, acetylcholine receptor, AF, atrial fibrillation; ALT, alanine transaminase, ANA, antinuclear antibodies, ARB, angiotensin receptor blockers; AST, aspartate aminotransferase, AVB, atrioventricular block; β1AR, beta-1-adrenergic receptor; BB, beta-blocker; BiPAP, bilevel positive airway pressure; BNP, brain natriuretic peptide; CAD, coronary artery disease; CC, calcium channel; CK, creatine kinase; CK-MB, creatine kinase—muscle/brain; CMR, cardiovascular magnetic resonance imaging; cN1A, cytosolic 5’-nucleotidase 1A; Cr, serum creatinine; CRP, C-reactive protein; CT, computer tomography; CTLA-4i, cytotoxic T-lymphocyte-associated protein 4 inhibitor; DLST, drug lymphocyte stimulation test; DM2, diabetes mellitus type 2; ECG, electrocardiography; eGFR, estimated glomerular filtration rate; EMB, endomyocardial biopsy; Eo, eosinophils; ESC, esophageal squamous cell carcinoma; GFAP, glial fibrillary acidic protein, Hb, hemoglobin; HCC, hepatocellular carcinoma; HCV, hepatitis C virus; HCQ, hydroxychloroquine; HMGCR, HMG-CoA reductase; HF, heart failure; HfpEF, heart failure with preserved ejection fraction; IABP, intra-aortic balloon pump; ICD, implantable cardioverter-defibrillator; IV, intravenous; IVIG, intravenous immune globulin; LCNEC, large cell neuroendocrine carcinoma; LDH, lactate dehydrogenasem; LyT, T lymphocytes; Mb, myoglobin; MPM, malignant pleural mesothelioma; mPSL, methylprednisolone; MG, myasthenia gravis; MMF, mycophenolate mofetil; MuSK, muscle-specific kinase; MV, mechanical ventilation; NC, nasal cannula; NG, nasogastric; NIPPV, nasal intermittent positive pressure ventilation; NOR90, nucleolus organizer region; NSCLC, non-small cell lung cancer; NT-pro-BNP, N-terminal pro hormone BNP; PO, per oral; PD-1i, programmed cell death 1 inhibitor; PD-L1i, programmed cell death ligand 1 inhibitor; PLEX, plasmapheresis; PLT, platelet count; PM/SCL, polymyositis/scleroderma; RCC, renal cell carcinoma; RYR, ryanodine receptor; SCS, systemic corticosteroid therapy; SJS, Stevens-Johnson syndrome; SRP, signal recognition particle; TEN, toxic epidermal necrolysis; Tn, troponin; TTS, Takotsubo syndrome, VA ECMO, veno-arterial extracorporeal membrane oxygenation, VT, ventricular tachicardia, WBC, white blood cells count.

**Table 2 ijms-23-10948-t002:** Case reports with ICI-associated pericarditis.

Study	Patient Characteristics	CV Side Effect	Symptoms	Diagnosis	Pre-Existent CVD	ICI	Type of Cancer	Pericarditis Onset	Concomitant Treatments	Pericarditis Treatment	Evolution	Concomitant AID
Khan, 2019	♂, 62	pericarditis	dyspnea	CT, echocardiogram, ECG, TnI, BNP	not mentioned	PD-1i	tonsillar cancer (squamous)	15 weeks	not mentioned	pericardiocentesis, prednisone	resolution	not mentioned
Arora, 2020	♂, 83	pericarditis	marked fatigue weakness chest pain orthopnea left eye ptosis	TnI, CK, TTE, brain MRI	Hypertension, HyperlipidemiaAtrial fibrillation	PD-1i	melanoma	1 month after first dose	not mentioned	colchicine and naproxen IV methylprednisolonePlasmapheresis Intubation	inability to reduce ventilatory support–transitioned to comfort measures	Hepatitis MG
de Almeida, 2018	♂, 69	pericarditis	dyspnea, tachycardia, low-grade fever	CT, echocardiogram	not mentioned	PD-1i	NSCLC (adenocarcinoma)	48 weeks	not mentioned	pericardiocentesis, prednisone	favorable	thyroiditis
Oristrell, 2018	♀, 55	pericarditis	pericardial chest pain	echocardiogram, ECG, TnI	not mentioned	PD-1i	ductal carcinoma of the left breast	30 weeks	not mentioned	Pericardiocentesis followed by pericardiectomy steroids	favorable	not mentioned
Zarogoulidis, 2017	♂, 60	pericarditis	not mentioned	not mentioned	not mentioned	PD-1i	NSCLC	17 weeks	not mentioned	pericardiocentesis mPSL	favorable	not mentioned
Öztürk, 2021	♂, 61	pericarditis	not mentioned	CT, echocardiogram, MRI	not mentioned	PD-1i	NSCLC	17 weeks	pemetrexed	not mentioned	not mentioned	not mentioned
Moriyama, 2021	♂, 58	pericarditis	fatigue, limb oedema, increased body weight	echocardiogram, cardiac CT, cardiac catheterization, EMB, MRI, ECG, TnT	not mentioned	PD-1i	NSCLC	77 weeks	not mentioned	prednisolone, furosemide, mPSL, infliximab	favorable	autoimmune hepatitis
Jacobs, 2021	♂, 54	pericarditis	chest pain, general malaise, dyspnea	echocardiogram, CT, MRI, ECG, TnI	not mentioned	PD-1i	NSCLC (adenocarcinoma)	5 weeks	carboplatin + pemetrexed +	mPSL	fatal	not mentioned
Dasanu, 2016	♀, 65	pericarditis	progressive dyspnea, chest discomfort	X-ray, CT, echocardiogram, ECG	not mentioned	PD-1i	melanoma (nodular type)	37 weeks	not mentioned	pericardiocentesis, mPSL	favorable	abnormal thyroid function, hepatitis, rash
Dhenin, 2019	♀, 79	pericarditis	intense thoracic pain, increasingwhen leaning, fatigue, general malaise	echocardiogram, ECG	hypertension	PD-1i	NSCLC (adenocarcinoma)	3 weeks	not mentioned	mPSL	favorable	rash, colitis, MG

Abbreviations: CT, computer tomography; ECG, electrocardiography; EMB, endomyocardial biopsy; IV, intravenous; LAFB, left anterior fascicular block; mPSL, methylprednisolone; MG, myasthenia gravis; MRI, magnetic resonance imaging; NSCLC, non-small cell lung cancer; PD-1i, programmed cell death 1 inhibitor; Tn, troponin.

**Table 3 ijms-23-10948-t003:** Case reports with ICI-associated arrhythmia.

Study	Patient Characteristics	CV Side Effect	Symptoms	Diagnosis	Pre-Existent CVD	ICI	Type of Cancer	Arrhythmias Onset	Concomitant Treatments	Arrhythmias Treatment	Evolution	Concomitant AID
Joseph, 2021	♂, 78	AF	not mentioned	ECG	hypertension	PD-1i	metastatic melanoma	35 weeks	not mentioned	TEE-guided cardioversion	favorable	not mentioned
	♂, 68	AF	not mentioned	ECG	hypertension	PD-1i	metastatic melanoma	14 weeks	not mentioned	self-limited AF (48 h)	favorable	not mentioned
	♂, 66	AF	not mentioned	ECG	hypertension	PD-1i	metastatic melanoma	2 weeks	not mentioned	beta-blockers	favorable	hyperthyroidism
	♂, 74	sinus bradycardia, AF with RVR	fatigue and dizziness	ECG	3 ablations for AF	PD-1i	metastatic melanoma	21 weeks	not mentioned	beta-blockers	favorable	not mentioned
Reddy, 2017	♂, 68	sinus tachycardia, 1st degree AVB, RBBB, LAFB followed bycomplete AVB)	fatigue, generalized malaise, weakness with ambulation NYHA IIB symptoms	ECG, Tn, CK-MB, echocardiogram, cardiac catheterization	not mentioned	PD-1i + CTLA-4i	metastatic sarcoma	2 weeks	not mentioned	high-dose IV steroids, temporary transvenous pacemaker, MMF	favorable	not mentioned
Giancaterino, 2020	♂, 88	ECGs–progression normal SR + with PAC- 3rd AVB (hospital day 5)	generalized weakness	ECG, Tn, CK-MB, TTE	not mentioned		invasive melanoma	first - 22 days prior		prednisone 40 mg daily nivolumab infusions were held IV methylprednisolone Infliximab dual-chamber pacemaker—day 10	decline clinically- VF- fatal	myositis
Behling, 2017	♂, 63	complete AVB (44 bpm)	worsening of a pre-existing dyspnea	ECG, echocardiogram, cardiac catheterization, myoglobin, Tn	hypertension	PD-1i	metastatic melanoma	3 weeks	not mentioned	temporary pacemaker, corticosteroids, oxygen therapy	fatal	not mentioned
Katsume, 2018	♂, 73	complete AVB (wide QRS complexes)	fatigue, faintness, syncope, palpitations	ECG, echocardiogram, cardiac catheterization, TnT, CK	not mentioned	PD-1i	metastatic NSCLC	2 weeks	not mentioned	IV steroids, pacemaker (temporary permanent)	favorable	not mentioned
Hsu, 2018	♂, 42	sinus bradycardia (40 bpm)	fatigue, dizziness, anorexia, hypotension	ECG, TnI	not mentioned	PD-1i	metastatic liver cancer	not mentioned	not mentioned	PO steroids	favorable	not mentioned
Pohl, 2020	♀, 61	new 1st degree AVB, QTc prolongation	not mentioned	ECG, echocardiogram	not mentioned	PD-1i	metastatic melanoma	4–12 weeks	not mentioned	not mentioned	not mentioned	not mentioned
Khan, 2020	♀, 67	Mobitz type 2 2nd degree AVB (30 bpm) complete AVB (22 bpm) after 3 h	asymptomatic	ECG, echocardiogram, TnT	hypertension, hyperlipidemia	PD-1i	metastatic NSCLC	3 weeks	not mentioned	dobutamine, pacemaker (temporary permanent)	favorable	not mentioned

Abbreviations: AF, atrial fibrillation; AVB, atrioventricular block; CK, creatine kinase; CK-MB, creatine kinase–muscle/brain; CTLA-4i, cytotoxic T-lymphocyte-associated protein 4 inhibitor; ECG, electrocardiography; IV, intravenous; LAFB, left anterior fascicular block; MMF, mycophenolate mofetil; NSCLC, non-small cell lung cancer; NYHA, New York Heart Association; PO, per oral; PD-1i, programmed cell death 1 inhibitor; RBBB, right bundle branch block; RVR, rapid ventricular response; TEE, transesophageal echocardiogram; Tn, troponin.

**Table 4 ijms-23-10948-t004:** Case reports with ICI-related cardiomyopathies.

Study	Patient	CV Side Effect	Symptoms	Diagnosis	Pre-Existent CVD	ICI	Type of Cancer	CVD Onset	CVD Treatment	Evolution
Serzan, (2021)	♀, 66	Takotsubo cardiomyopathy	Exertional dyspnea, generalized pain	TNI, ECG, TTE, CT angiography, RV catheterization, EMB, CMR	not mentioned	CTLA-4i + PD-1i	Choroidal melanoma	16 weeks	Metoprolol	Favorable
Oldfield, (2021)	♂, 76	Takotsubo cardiomyopathy, diabetic ketoacidosis	Chest pain, diaphoresis	ECG, TNI, Coronary angiography, TTE, CMR	T2DM, dyslipidemia, hypertension	CTLA-4i + PD-1i	Melanoma	4 days	Aspirin, bisoprolol, ramipril	Favorable
Schwab, (2018)	♂, 69	Takotsubo cardiomyopathy	Chest pain, shortness of breath	Coronary angiography, TTE, CMR	not mentioned	PD-1i and CTLA-4i + PD-1i	Squamous cell cancer of the lower lip	7 cycles	Heart failure treatment, prednisolone	Favorable
Ederhy, (2018)	♂, 41	Takotsubo-like syndrome	not mentioned	ECG, TNI, TTE, Coronary angiography, CMR	not mentioned	CTLA-4i + PD-1i	melanoma	5 days	IV mPSL	Favorable
	♂, 77	Takotsubo-like syndrome	not mentioned	ECG, CMR, TNI, Coronary angiography	not mentioned	CTLA-4i + PD-1i	esophageal melanoma	2 cycles of ipilimumab + nivolumab and 1 cycle of nivolumab	mPSL, ACEi, beta blockers	Favorable
Tan, (2020)	♂, 62	Takotsubo cardiomyopathy	Chest pain, nausea, vomiting	ECG, Coronary angiography, TNI, NT-proBNP, TTE, CMR	not mentioned	PD-1i	HCC	3 weeks	IV mPSL, Prednisone	Favorable
Al-Obaidi, (2020)	♀, 52	Acute HF	Dyspnea on exertion, angina-like chest pain, lower extremity oedema	TTE, chest X-ray, CT angiography, ECG	not mentioned	CTLA-4i + PD-1i	NSCLC	1 year	IV mPSL, oral prednisone	Favorable
Khan, (2020)	♀, 57	Atypical Takotsubo cardiomyopathy	Chest pain, palpitations, tachypnoea, tachycardia	Chest X-ray, ECG, TNI, TTE, Coronary angiography	not mentioned	PD-L-1i	NSCLC (adenocarcinoma)	2 weeks following 4th cycle	Guideline-directed HF treatment	Favorable
Samejima, (2020)	♂, 79	Acute HF	Dyspnea	Chest X-ray, CT, ECG, CRP, CK, CK-MB, TnI, BNP, TTE, Coronary angiography, EMB	not mentioned	PD-1i	NSCLC	20 days	Furosemide, dopamine, tolvaptan, bisoprolol, spironolactone, enalapril	Favorable
Roth, (2016)	♂,60	Left Ventricular Dysfunction	heart palpitations	ECG, TTE, pharmacologic stress test	hypertension, anxiety, and Raynaud syndrome	CTLA-4i	BRAFwild-type stage IIIA (T2, N1a) melanoma	4 cycles + another 4 cycles (liver metastasis)–after 4 months	beta blockers, ACEI	Favorable
Andersen, (2016)	♀, 56	Apical takotsubo syndrome	chest pain after severe episode of abdominal cramping with diarrhea	ECG, highly sensitive troponin Chest X-ray, TTE, Coronary angiography	no cardiac risk factors. No CVD	PD-1i	breast carcinoma	3 weeks	ACEi, beta blockers	Favorable

Abbreviations: ACEi, angiotensin-converting enzyme inhibitor; ACS, acute coronary syndrome; BNP, brain natriuretic peptide; CCB, calcium channel blocker; CTLA-4i, cytotoxic T-lymphocyte-associated protein 4 inhibitor; cTnI, cardiac troponin I; CK, creatine kinase; CK-MB, creatine kinase-MB; CMR, Cardiac MRI; CRP, C-reactive protein; ECG, electrocardiogram; HF, heart failure; HCC, hepatocellular carcinoma; mPSL, methylprednisolone; NSCLC, non-small cell lung cancer; NT-proBNP, N-terminal pro-brain natriuretic peptide; PD-1i, programmed cell death protein 1 inhibitor; PD-L1i, programmed death-ligand 1 inhibitor; RV, right ventricle; T2DM, Type 2 diabetes mellitus; TTE, transthoracic echocardiogram.

**Table 5 ijms-23-10948-t005:** Case reports with ICI-related acute coronary syndrome.

Study	Patient	CV Side Effect	Symptoms	Diagnosis	Preexistent CVD	ICI	Type of Cancer	CVD Onset	CVD Treatment	Evolution
Arora, 2020	♂,69	ACS (NSTEMI)	diffuse body pain weakness	ECG, CK-MB, cTNI, TTE Coronary angiography	CKD, Hypertension, Hyperlipidemia Type 2 DM CAD	PD-1i	metastatic urothelial carcinoma	cycle day 2	IV Steroids, MMF	transition to comfort measures
Cheng, (2021)	♀, 87	ACS (NSTEMI)	Chest pain and dyspnea	ECG, CK-MB, cTNI, CRP, Coronary angiography	Hypertension, 3-vessel CAD	PD-1i	NSCLC (adenocarcinoma)	2 days	PCI + DES	Favorable
Tomita, (2017)	♂, 61	ACS (NSTEMI)	not mentioned	CK, CK-MB, cTNI, Coronary angiography, OCT	Dyslipidemia	PD-1i	NSCLC (adenocarcinoma)	11th cycle	PCI + DES; thrombus aspiration	Favorable
Kwan, (2019)	♀, 71	ACS (NSTEMI)	Chest pain	cTNI, ECG, Coronary angiography	Hypertension, T2DM, peripheral artery disease	PD-1i	Giant cell tumor of the bone	2 Years	First ACS: atherectomy of the LAD with 3 x DES, aspirin + clopidogrel, atorvastatin; Second ACS: DES, DAPT	Favorable
Cancela-Díez, (2019)	♂, 79	ACS (STEMI)	Chest pain, oppression and dyspnea.	TNI, TTE, ECG, Coronary angiography	Infrarenal abdominal aortic aneurysm	PD-1i	NSCLC (epidermoid)	10 days after the last cycle (10th)	PCI + DES, aspirin + clopidogrel, nitro-glycerine, beta blockers, enalapril	Favorable
Masson, (2020)	♂, 62	ACS (NSTEMI)	Chest pain	ECG, TNI, BNP, TTE, Coronary angiography	T2DM, multi-vessel CAD, STEMI	PD-1i	Melanoma	1 week after cycle 4 of Nivolumab therapy	CABG	Favorable
Cautela, (2020)	♀, 52	ACS (NSTEMI)	Chest pain	ECG, TNT, NT-proBNP, TTE, Coronary angiography, CMR	not mentioned	PD-1i	NSCLC (?)	5 days	Methylprednisolone prednisolone; aspirin + clopidogrel, statins	Fatal (due to refractory shock)
Otsu, (2020)	♂, 57	Vasospastic angina	Rest angina	ECG, Coronary angiography	not mentioned	PD-1i	Renal cell carcinoma	4 weeks	CCB, nitrates	Favorable
Kumamato, (2022)	♀, 54	Vasospastic angina	chest pain at rest for 2 months	ECG, Chest radiography, TTE, Gadolinium enhanced cardiac MRI cardiac catheterization–coronary vasospasm provoked by ergonovine	not mentioned	PD-1i	Hypopharyngeal cancer	21 months	Benidipine 8 mg	Favorable
Guo, (2022)	♂, 6o	Coronary Artery Spasm ventricular tachycardia	1-week history of chest tightness and palpitation	ECG Thyroid function Holter ECG TTE	acute coronary syndrome 8 months ago–complete revascularization with stents	PD-1i	Metastatic liver cancer	Pre-evaluation of 3^rd^ dose	isosorbide mononitrate and diltiazem sedative drug aspirin, clopidogrel, and atorvastatin	1 week later–discharged

Abbreviations: ACS, acute coronary syndrome; BNP, brain natriuretic peptide; CABG, coronary artery bypass grafting; CAD, coronary artery disease; CCB, calcium channel blockers; cTnI, cardiac troponin I; CK, creatine kinase; CK-MB, creatine kinase-MB; CMR, Cardiac MRI; CRP, C-reactive protein; DAPT, dual antiplatelet therapy; DES, drug-eluting stent; ECG, electrocardiogram; LAD, left anterior descending artery; NSCLC, non-small cell lung cancer; NSTEMI, non-ST segment elevation myocardial infarction; NT-proBNP, N-terminal pro-brain natriuretic peptide; OCT, Optical coherence tomography; PCI, percutaneous coronary intervention; PD-1i, programmed cell death protein 1 inhibitor; STEMI, ST segment elevation myocardial infarction; T2DM, type 2 diabetes mellitus; TTE, transthoracic echocardiogram.

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
