# Peer review of "Cardiac Toxicity Associated with Immune Checkpoint Inhibitors: A Systematic Review"

_ijms, 2022, doi:10.3390/ijms231810948_

Round 1
Reviewer 1 Report
The authors provide a review of cardiac toxicity associated with immune check point inhibitors (ICI). This aspect is relevant and not so comprehensively addressed in the literature, which warrants publication. Some minor changes should be made before the review can be accepted
- Unless it is a recommendation from the editor, I would suggest not to follow a "materials and methods" and "results" plan but rather to prefer one paragraph per message.
- page 3, paragraph 3.1: the authors mention that 118 studies were included although 101 articles are mentioned in Figure 1. These numbers are not clear to me and seem discordant
- Table 1: The table highlights the fact, as also mentioned in the text, that medical history is very rarely available in studies. What about the smoking status of the included patients, is it mentioned in the indicated studies?
- although the mechanisms of ICI-related cardiac toxicity are not yet fully understood, this aspect is very quickly mentioned and should be a little more developed.
- Are there any data regarding novel immune check point inhibitors used in the clinics (meaning targeting different immune check point than PD-1/ PD-L1/CTLA4) and cardiac toxicity. What about combination treatments (again, besides PD1/PD-L1 and CTLA4)? This would be an interesting aspect to address at the end of the article.
Author Response
First of all, we appreciate the time and effort you dedicated to providing feedback on our manuscript. We are truly grateful for your insightful comments, as well as for your improvement suggestions, which we found quite helpful.
We have incorporated all your recommendations in our revised paper. Consequently, you will find both content and structural changes in the manuscript. Thank you for your very professional recommendations. We are confident that the revised version of our paper is improved, with a better flow, coherence, and narrative.
In the revised manuscript, we highlighted all significant additions to and alterations of the initial version. See below, point-by-point, our responses to each of your comments and concerns.
Best regards,
Authors
- Unless it is a recommendation from the editor, I would suggest not to follow a "materials and methods" and "results" plan but rather to prefer one paragraph per message.
We have respected the authors’ instructions; therefore, for the moment, we have let the proposed article structure. But, if the editor finds it appropriate to change the presentation, it’s not a problem for us.
- page 3, paragraph 3.1: the authors mention that 118 studies were included although 101 articles are mentioned in Figure 1. These numbers are not clear to me and seem discordant
We have checked – we have corrected the date. Supplementary, we have added another found studies (from Embase, Cochrane).
We have re-done the Flow diagram of the paper selection processes
- Table 1: The table highlights the fact, as also mentioned in the text, that medical history is very rarely available in studies. What about the smoking status of the included patients, is it mentioned in the indicated studies?
Thank you very much for your very good suggestion to quantify smoking’s effect on immune checkpoint inhibitors cardiotoxicity.
As it is already known, there are some data about the influence of smoking on the efficacy of immune checkpoint inhibitors – some recently published papers show that the efficacy of ICIs in patients with smoking history is seemingly superior to patients without smoking history (DOI: 10.21037/jtd-20-1953, DOI: 10.1080/0284186X.2019.1670354)
Unfortunately, no reported data are presenting the relationship between smoking habit and toxicity (in general); studies are necessary. To our best knowledge, no data has been now published on this subject – but it is a very good idea.
- although the mechanisms of ICI-related cardiac toxicity are not yet fully understood, this aspect is very quickly mentioned and should be a little more developed.
We have developed the topic and also, and we have added a figure for a better subject understanding.
- Are there any data regarding novel immune check point inhibitors used in the clinics (meaning targeting different immune check point than PD-1/ PD-L1/CTLA4) and cardiac toxicity. What about combination treatments (again, besides PD1/PD-L1 and CTLA4)? This would be an interesting aspect to address at the end of the article.
Thank you very much for the set forth idea – we have completed the article with a short paragraph about the relationship between novel immune check point inhibitors and cardiotoxicity.
Reviewer 2 Report
This systematic review summarized recent publications on immune checkpoint inhibitors-induced cardiac toxicities and highlight the therapeutical approach and the evolution in the selected cases. Immune checkpoint inhibitors-induced cardiac toxicities have been reviewed in some studies, but new publications are included in this paper. Study design and search strategy are quite appropriate for the scope of the manuscript.Although there are several typos and the narrative is a kind of fragmentary, I just abide by scientific soundness. However, there are some issues and questions that should be addressed in the study.
1- Please indicate why you need to perform this review and how it differs from others , although there are many reviews on this subject. ( https://doi.org/10.3390/cancers13205218, https://doi.org/10.1007/s11912-021-01070-6, https://doi.org/10.1016/j.ejca.2021.01.043, https://doi.org/10.3389/fphar.2019.01350
)
2- In introduction and discussion section the authors should discuss more extensively about the pathophysiology and mechanism of Immune Checkpoint Inhibitor-Associated Cardiotoxicity. Perhaps add a figure illustrating the underlying mechanism and cardiac toxicity associated with an immune checkpoint inhibitor (ICI)
3- I think it would be better to add immunosuppressive therapy and ICI rechallenge parameters to the tables.
4- Why didn't you use the Embase and medline databases with Pubmed, which has the opportunity to find more studies while doing a systematic search?
5- Please clearly indicate the systematic search dates in Methods.
6- “Between 2017 and January 2022 there were four clinical trials that reported cardiac
irAEs following the administration of ICIs” - Please check this sentence for more clinical trials ( https://doi.org/10.1016/j.ejca.2021.01.043)
Author Response
First of all, we appreciate the time and effort you dedicated to providing feedback on our manuscript. We are truly grateful for your insightful comments, as well as for your improvement suggestions, which we found quite helpful.
We have incorporated all your recommendations in our revised paper. Consequently, you will find both content and structural changes in the manuscript. Thank you for your very professional recommendations. We are confident that the revised version of our paper is improved, with a better flow, coherence, and narrative.
In the revised manuscript, we highlighted all significant additions to and alterations of the initial version. See below, point-by-point, our responses to each of your comments and concerns.
Best regards,
Authors
Comments and Suggestions for Authors
- Please indicate why you need to perform this review and how it differs from others , although there are many reviews on this subject. ( https://doi.org/10.3390/cancers13205218, https://doi.org/10.1007/s11912-021-01070-6, https://doi.org/10.1016/j.ejca.2021.01.043, https://doi.org/10.3389/fphar.2019.01350)
Immune checkpoint inhibitors (ICIs) are an important recent advancement in the field of cancer treatment. Their use is in relationship with a lot of side effects.
Yes, we agree with you that other authors have had similar research ideas. At the same time, we need (for better daily practice) to put together data and publish as many papers on this topic.
We have tried to bring into attention some useful details – to offer data for adequate diagnosis and treatment.
2- In introduction and discussion section the authors should discuss more extensively about the pathophysiology and mechanism of Immune Checkpoint Inhibitor-Associated Cardiotoxicity. Perhaps add a figure illustrating the underlying mechanism and cardiac toxicity associated with an immune checkpoint inhibitor (ICI)
Thank you very much for the very good suggestion. We have completed the article, and we added a figure illustrating the underlying mechanisms
3- I think it would be better to add immunosuppressive therapy and ICI rechallenge parameters to the tables.
Yes, it is very important what you have suggested. Unfortunately, the tables are already so big – so it is difficult to give some more details. We have tried, to put in every line of data about the treatment.
4- Why didn't you use the Embase and medline databases with Pubmed, which has the opportunity to find more studies while doing a systematic search?
As you suggested, we have extended the search also to Embase and Medline databases. We have found more interesting papers – we have added them and we have up-dated the bibliography. 5- Please clearly indicate the systematic search dates in Methods.
We have done
6- “Between 2017 and January 2022 there were four clinical trials that reported cardiac irAEs following the administration of ICIs” - Please check this sentence for more clinical trials ( https://doi.org/10.1016/j.ejca.2021.01.043)
Thank you – it was our mistake. We have taken into consideration the suggestion – we have read the suggested article and followed the presented data.
Reviewer 3 Report
Cardiac Toxicity Associated to Immune Checkpoint Inhibitors: A Systematic Review by Angela Cozma is the summary of relevant literature in the field and may be published without any reservation
Author Response
We appreciate the time and effort you dedicated to providing feedback on our manuscript.
We are truly grateful for the appreciation.
Best regards,
Authors
Reviewer 4 Report
The review presents data associating Immune checkpoint inhibitors (ICIs) and cardiac adverse effects. The work sounds more like a description of data and facts than a "critical review of the theme." The authors should include their thoughts and critically discuss the results presented.
For instance, it is critical not to have complete patient history documentation. The lack of patient history is a significant drawback because it implies not having enough information to analyze the results critically.
The author should include a "limitation section".
For instance, are the patients obese? Do they have metabolic syndrome or diabetes? Do they have any preexisting metabolic or inflammatory conditions?
Author Response
First of all, we appreciate the time and effort you dedicated to providing feedback on our manuscript. We are truly grateful for your insightful comments, as well as for your improvement suggestions, which we found quite helpful.
We have incorporated all your recommendations in our revised paper. Consequently, you will find both content and structural changes in the manuscript. Thank you for your very professional recommendations. We are confident that the revised version of our paper is improved, with a better flow, coherence, and narrative.
In the revised manuscript, we highlighted all significant additions to and alterations of the initial version. See below, point-by-point, our responses to each of your comments and concerns.
Best regards,
Authors
The review presents data associating Immune checkpoint inhibitors (ICIs) and cardiac adverse effects. The work sounds more like a description of data and facts than a "critical review of the theme." The authors should include their thoughts and critically discuss the results presented.
We have tried to comment and discussed the data, but in a short manner – the article is already an extensive presentation of cardiotoxicity associated with ICI use.
For instance, it is critical not to have complete patient history documentation. The lack of patient history is a significant drawback because it implies not having enough information to analyze the results critically.
We completely agree with you. But unfortunately, our article is based on published clinical cases – not always containing all the necessary data. This type of article is just the beginning of the following one.
For the moment, we intend to see if some patterns for cardiotoxicity / useful treatment can be highlighted.
The author should include a "limitation section".
We have done. Thank you for the suggestion.
Round 2
Reviewer 2 Report
I am satisfied that the authors have addressed all of my previous concerns about the article. It is now much improved and I feel that it is now suitable for publication.
Reviewer 4 Report
The authors have revised the manuscript accordingly. Despite the lack of some information about the patients, they added this to the limitation section.